# How Does Paired Assistance to Disaster-Affected Areas (PADAA) Contribute to Economic Sustainability? A Qualitative Analysis of Wenchuan County

**Xiaojun Zhang [1,2]** and **Zhiqiang Wang [3,]\***

1   School of Economics & Management, Fuzhou University, Fuzhou 350108, China
2   Institute for Risk and Disaster Reduction, University College London, London WC1E 6BT, UK
3   School of Public Administration, South China University of Technology, Guangzhou 510641, China
*   Correspondence: pawangzhiqiang@scut.edu.cn

**Abstract:** There is a high risk of an economic downturn after the end of reconstruction efforts following natural hazards. High levels of external assistance can sometimes weaken local autonomy and self-sufficiency, creating the pre-conditions for a "forgotten phase". However, through the three-year Paired Assistance to Disaster-Affected Areas project (PADAA), the economy of Wenchuan County in China recovered to its pre-earthquake levels within two years and has shown clear signs of economic sustainability. Through a qualitative research approach based on the analysis of expert interviews, secondary data, and relevant documentation, this study discusses the phases of the reconstruction process following the Wenchuan earthquake, and the factors behind the success of the PADAA process in enabling economic sustainability. Some of the identified factors include: (1) the reshaping of local livelihoods and economic structure through a large number of investments in public infrastructure; (2) knowledge acquisition, self-adjustment, and the ability to meet the needs of a new economy and social development through institutional reform and openness; (3) increasing amounts of attracted investments and the development of sustainable industrial structures through the improvement of the local government's economic governance.

**Keywords:** paired assistance; long-term recovery; Wenchuan Earthquake; China

## 1. Introduction

Natural disasters can have serious impacts on national, regional, and local economies [1]. There have been many examples of economic downturn after natural disasters [2–4], with economic systems sometimes underperforming in the long-term, or even entering a "forgotten phase" [5]. This "forgotten phase" tends to follow the initial rapid economic growth in the first few years after the disaster, as reconstruction efforts decelerate [5]. This often manifests in declining levels of employment and income that sometimes fall below pre-disaster levels [6]. For instance, empirical evidence from the Japanese Great Hanshin-Awaji earthquake (i.e., Kobe earthquake) shows that the three-year emergency recovery plan and the 10-year reconstruction plan could not prevent the long-term decline in the gross regional product (GRP) following the earthquake [7]. Actually, the initial reconstruction efforts led to a short-term economic upturn, followed by a period during which economic activity was well below the pre-earthquake levels. For instance, the GRP of the affected areas started rising seven years after the earthquake, but it did not reach its pre-earthquake levels until eleven years after the earthquake [6]. Scholars have suggested that reconstruction efforts can cause this economic "boom and bust" cycle

if there are no alternative sources of employment when the reconstruction efforts have either been completed or have no more economic resources [8].

Sustainability in the context of natural disasters implies the capacity of affected areas to recover in the long-term by effectively utilizing their internal and external resources [9,10]. In such contexts, sustainability reflects the ability of localities to "tolerate—and overcome—damage, diminished productivity, and reduced quality of life from an extreme event" [10]. Many scholars have adopted and applied the concept of sustainability to study recovery and reconstruction processes following natural disasters [11–13]. However, sustainability not only entails the ability to cope with the immediate impacts of natural disasters, but also ensures that affected regions sustain their economic growth in the short- and long-term [14]. Thus, in the context of natural disasters, economic sustainability involves the ability of the local economy in disaster-affected regions to support a defined level of economic production in the long-term, avoiding the forgotten phase [15].

The most important criteria pertaining to economic sustainability in disaster-affected areas usually relate to employment opportunities, residents' livelihood, industrial restoration, and positive economic impact [16]. Many different approaches have been deployed around the world to ensure the sustainable economic development of disaster-affected regions following natural disasters including land use planning (and related policies) [10,17,18], capital stock preservation and upgrading [19,20], infrastructure reconstruction [21], humanitarian efforts [22], and the adoption of new technology [20], among others.

One of the different disaster reconstruction models is the Paired Assistance to Disaster-Affected Areas (PADAA) approach, commonly implemented in China. The PADAA model is a type of national aid system that assigns an economically prosperous region to offer economic support to a disaster-affected area [23,24]. The economic support is directed towards reconstruction efforts and ultimately facilitates recovery under the arrangement and overseeing of the central government [25]. The PADAA models entails a comprehensive approach towards funding, policy support and technical assistance across different aspects such as urban/rural housing, town/rural construction, public services, infrastructure, industry, disaster prevention and mitigation, and ecological/environmental restoration [26]. According to the regulations of the central government, the annual investment of the donor province must be no less than 1% of its financial income in the previous year and should be provided for three years after the disaster [27,28]. Thus, the primary objectives of the PADAA process is to assist in the recovery of disaster-affected areas, and to provide technical guidance with the close cooperation of local governments through the facilitation and overseeing of the national government.

The PADAA system was deployed in China on many occasions, such as after major disasters and crises. Some examples include the emergency support after the 1998 floods [23], poverty alleviation efforts [29], and the "Western Development Strategy" (the "Western Development Strategy" is a policy of the Chinese government launched to assist in the development of western regions of China) [30], among others, to establish links between two local governments [23]. The 2008 Wenchuan earthquake was one of the major disasters for which the PADAA model was mobilised (see Section 2.1). Municipalities in Guangdong Province assisted areas in Wenchuan County through a three-year PADAA project to allow the county to recover to its pre-disaster state. This was achieved within two years [1], with the GRP maintaining a steady increase since the official end of the PADAA process. Approaches similar to the PADAA model were mobilized during the reconstruction and recovery process of the Japanese Kobe and Tohoku earthquakes, and Italian earthquakes, among others [31–33].

Conventional research on PADAA implementation for disaster recovery in China has mainly focused on the operational mechanisms in place before the completion of the associated program [23,24]. Thus, little is understood about the economic outcomes of PADAA processes, especially in terms of what happens following the official end of the assistance period. Few scholars have paid attention to the actual effects of PADAA processes on long-term recovery and sustainability, even raising doubts about the sustainability of the program itself [34]. While the Wenchuan earthquake recovery process received

considerable attention due to its perceived success [5,35,36], it is not clear how the institutional aspects of the PADAA process contributed to this success, especially as pertaining to economic sustainability.

To fill these knowledge gaps, this article uses a qualitative research approach to explore the factors and mechanisms that enabled the PADAA process to effectively assist disaster-affected areas in Wenchuan County to avoid a "forgotten phase" and display characteristics of economic sustainability. This paper analyzes the recovery of Wenchuan County from the perspective of key economic impacts and underlying causes for concern. To achieve this, we reconstruct the main phases of the PADAA process and identify key economic changes, using thematic analysis to explore the factors that enhanced the economic sustainability of Wenchuan following the earthquake. We synthesize these insights to propose a theoretical framework to guide external assistance and economic recovery following natural disasters by leveraging the role of the cooperation between local governments.

Section 2 outlines the main aspects of the PADAA process followed after the Wenchuan earthquake and the data collection and data analysis methods. Section 3 presents the main results divided across the three stages of the PADAA process and its aftermath. Section 4 discusses the main implications of this study, especially as it relates to the operational aspects of the PADAA system and its outcomes for economic sustainability.

## 2. Methods

### 2.1. Study Site

The Wenchuan earthquake (512 earthquake) had a surface wave magnitude of 8.0 and occurred in Wenchuan County of Sichuan Province, China, on May 12, 2008. The disaster seriously impacted the economy, infrastructure, and local communities in more than ten counties in the Sichuan Province [1]. One of the most affected areas was Wenchuan County, which was the epicenter of the earthquake. According to incomplete statistics (see Table 1), the earthquake caused 15,941 deaths and 34,583 injuries in Wenchuan County alone. The earthquake also had a significant economic impact, reducing the total gross regional product (GRP) by approximately 47% in its aftermath (i.e., from 0.42 billion USD in 2008, to 0.22 billion USD in 2008) [37].

**Table 1.** Basic characteristics of the Wenchuan earthquake.

| Items | Wenchuan |
|---|---|
| Gross regional product (GRP) (2007) | 0.42 billion USD |
| GRP (2008) | 0.20 billion USD |
| Deaths | 15,941 (person) |
| Injured people | 34,583 (person) |

Source: National Bureau of Statistics of the People's Republic of China, Sichuan Provincial Bureau of Statistics.

After the earthquake, the central government of China identified 19 provinces or municipalities directly under its control in more prosperous areas of eastern and central China that could assist the Sichuan Province. Guangdong Province was chosen to assist the affected Sichuan Province (Figure 1). Guangdong is a highly developed coastal province and is part of China's reform and opening up process. Since 1989, Guangdong has had the largest GRP in China, accounting for one-eighth of the country's total national gross domestic product (GDP), having reached the same economic level as middle-income and moderately-developed countries.

The system of bureaucracy between donors and aid recipients in this PADAA process was similar to a "top-down" tree structure [23], pairing administrative regional divisions in Guangdong and Sichuan from top to bottom, i.e., from the province down to the city, district, county, and town level. Guangdong was scheduled to help Wenchuan reconstruct, and it delegated tasks downward to the prefecture level (Figure 2, Table 2). In other words, one city district was arranged to help one village in the town.

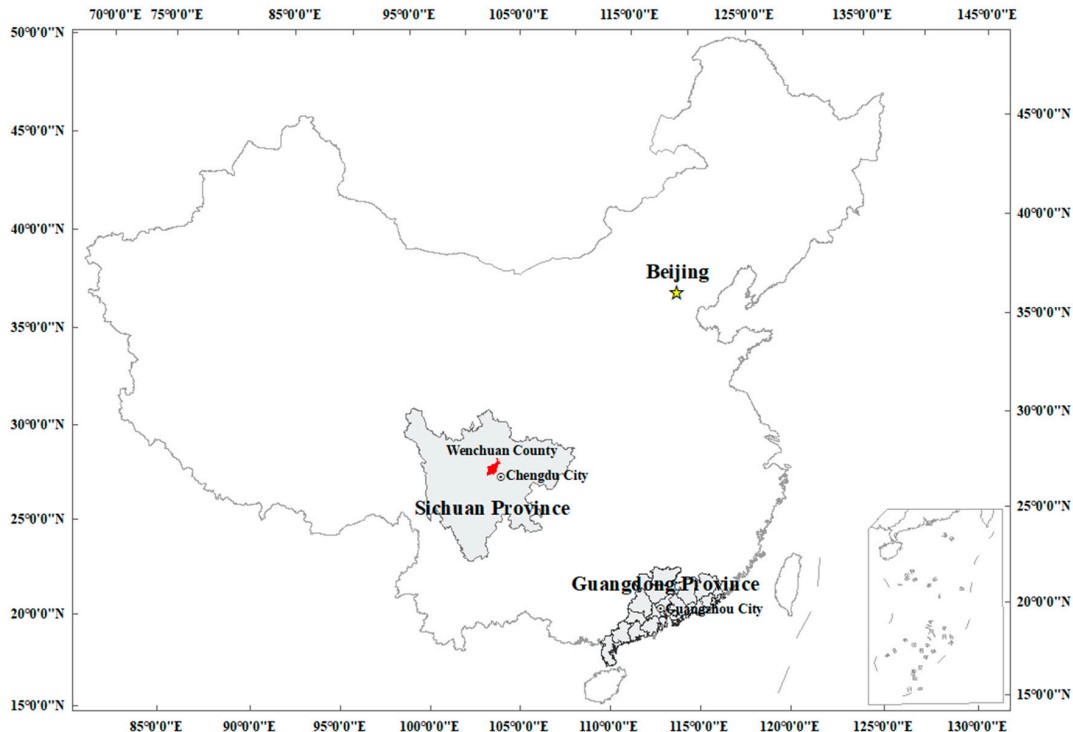

**Figure 1.** Study site: Wenchuan County (in red) and Guangdong Province.

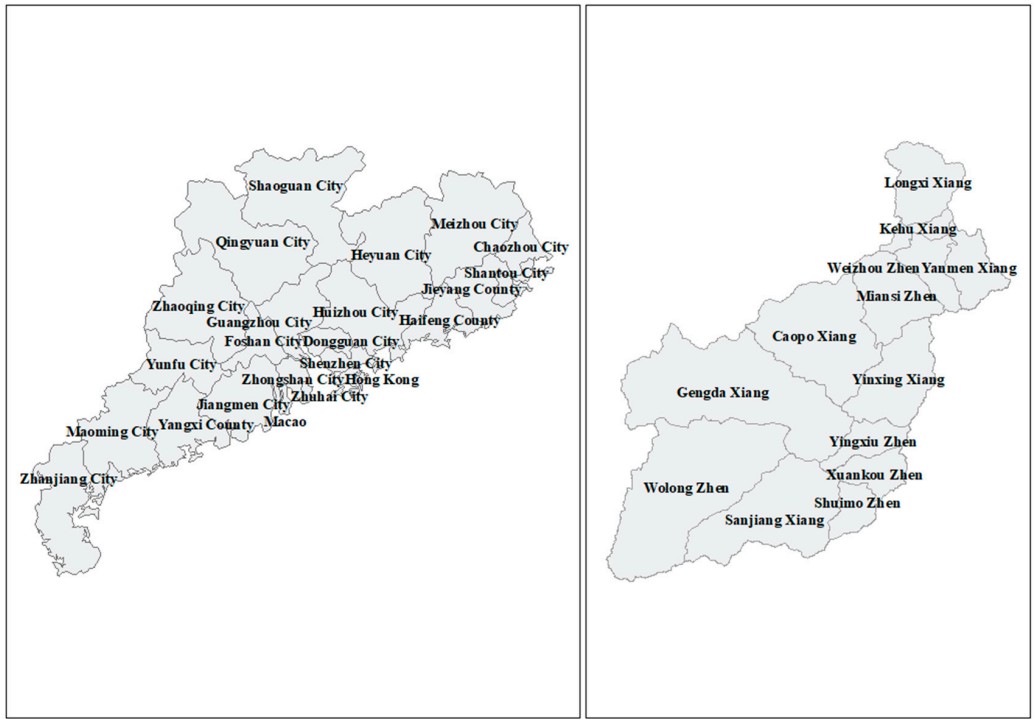

**Figure 2.** Participating cities in the "Guangdong-Wenchuan" paired assistance relations.

Districts and counties, large state-owned enterprises, and public institutions requested their residents, citizens, and employees to voluntarily participate in specific aid projects. The PADAA process started immediately after the earthquake in 2008 and lasted for three years until 2011. Overall, the Guangdong–Wenchuan paired assistance had an extremely large-scale impact, in terms of disaster and post-disaster recovery mobilization, ranking first in the contribution scale of the paired assistance policy [38–40].

**Table 2.** "Guangdong–Wenchuan" prefecture-level city paired assistance relations.

| Donors (Guangdong) | Aid Recipients (Wenchuan) |
| --- | --- |
| Guangzhou City | Weizhou Zhen (town) |
| Zhuhai City | Mianzhu Zhen (town) |
| Shantou City | Caopo Xiang (town) |
| Foshan City | Shuimo Zhen (town) |
| Huizhou City | Sanjiang Xiang (town) |
| Dongguan City | Yingxiu Zhen (town) |
| Zhongshan City | Xuankou Zhen (town) |
| Jiangmen City | Yanmen Xiang (town) |
| Zhanjiang City | Longxi Xiang (town) |
| Maoming City | Yinxing Xiang (town) |
| Zhaoqing City | Keku Xiang (town) |
| Chaozhou City | Gengda Xiang (town) |
| Jieyang City | Wolong Zhen (town) |

## 2.2. Data Collection and Analysis

Qualitative research methods were used to investigate the factors and mechanisms that enabled or hindered the success of the Wenchuan PADAA. A grounded theory approach was used to enable the empirical analysis of data collected during three main phases: (a) during PADAA (2009–2010), (b) immediately after PADAA (2012–2015), and (c) a few years after the end of PADAA (three continuous months in 2017–2018). These phases were selected to represent different stages of the reconstruction stage as mediated by the PADAA process.

We used multiple methods to collect data during the study periods. These included informal interviews, participant observation, semi-structured expert interviews, secondary data, and documentary/archival records such as internal reports, newspaper articles, official websites, media coverage, files, and policy papers (Tables 3 and 4). Relying on multiple data sources can improve the reliability and validity of the case study by allowing data complementarity and cross validation.

We conducted in-depth expert semi-structured interviews with officials from different Wenchuan County departments, the Development and Reform Commission of Guangdong Province (GDDRC) and the Sichuan Province's office in Guangzhou, as well as other local officials, leaders of social organizations and enterprises, survivors and individuals who participated in the PADAA processes (Table 3). The main aspects covered in these interviews included information about (a) the PADAA projects and overall processes; (b) the role of PADAA in transforming economic structures; (c) the effects of PADAA on local community livelihoods; (d) the factors influencing economic recovery; and (e) the effects of PADAA on the economic sustainability of Wenchuan County.

Informal interviews were conducted with officials and residents in disaster-affected areas, where the respondents outlined aspects of the economic recovery based on their previous knowledge and experience. The outline of the informal interviews was similar to that of the semi-structured interviews. However, informal interviews were conducted through casual conversations.

Participant and site observations were conducted by one of the authors who served as an evaluator of the recovery of Wenchuan County, checking PADAA achievements in 2018, and especially, the current operation of recovery projects. Furthermore, one of the authors was a consultant of the Guangdong Province emergency management and was invited to assess the earthquake damage and the types of reconstruction projects undertaken.

**Table 3.** Role and affiliation of expert interviewee participants.

| No. | Role of Interviewee | Organization/Department | Code | Note |
|---|---|---|---|---|
| 1 | President | Yingxiu Central Hospital of Health | A1-YX020 | Paired Assistance to Disaster-Affected Areas (PADAA) project |
| 2 | Vice President | Wenchuan No. 1 Middle School | A4-JG001 | PADAA project |
| 3 | Vice Director | Wenchuan County Civil Affairs Bureau | A1-WC003 | Member of Wenchuan County working group on receiving PADAA from Guangdong Province |
| 4 | Chairman | Jiangmen Working Group | A1-CP007 | City-level working group from Guangdong Province |
| 5 | Local Resident | Weizhou Town | A5-WZ005 | |
| 6 | Local Resident | Yingxiu Town | A5-YX003 | |
| 7 | Local Resident | Keku Town | A5-KK002 | |
| 8 | Director | Wenchuan Economic Commerce and Information Technology Bureau | A1-JX004 | |
| 9 | Director | Weizhou Town | A1-WZ001 | |
| 10 | Vice Chairman | China Post in Wenchuan | A5-WZ012 | |
| 11 | Chairman | Wenchuan Datong Social Worker Station | A2-DT007 | First social worker organization in Wenchuan in terms of PADAA support |
| 12 | Professor | Guangdong University of Technology | A2-DT008 | Sponsor of the Wenchuan Datong social worker station |
| 13 | President | Wenchuan County | A2-SG021 | Worked in Wenchuan County for more than 12 years |
| 14 | Vice Chairman | Guangdong Social Organization Administration | A1-SHZZ013 | Member of working group from Guangdong Province |

**Table 4.** Coding source and data classification.

| Type | Source | Coding | | | | |
|---|---|---|---|---|---|---|
| | | Government | /Social Organizations | Private Sector | PADAA Participants | Local Residents |
| Primary data | Expert interviews | A1 | A2 | A3 | A4 | A5 |
| | Informal interviews | B1 | B2 | B3 | B4 | B5 |
| | Site observations | C1 | C2 | C3 | C4 | C5 |
| Secondary data | Online resources | a1 | a2 | a3 | a4 | a5 |
| | Documents | b1 | b2 | b3 | b4 | b5 |

Note: PADAA participants denote stakeholders directly involved in the PADAA project, representing the government, social organizations, and the private sector.

Table 3 outlines the expert interviews included in this study and Table 4 presents the different data sources. It should be noted that many more interviews were conducted during the three project phases outlined above.

When analyzing interruptions caused by natural disasters, an important starting point is the reconstruction of the event, as a means of better understanding the physical and social aspects of the event as well as people–people and people–object relationships [41]. In this phase, we tried to reconstruct the PADAA and its aftermath using the aforementioned data that were collected across the three phases. We used thematic analysis to explore the factors that enabled the sustainable economic recovery of Wenchuan's economy following the PADAA process.

NVivo 12 software was used to analyze the interviews and secondary data. We used deductive theme analysis comprising six stages to identify meaningful patterns within the information. These

stages included (a) data familiarization, (b) generation of initial codes, (c) theme searching amongst codes, (d) theme reviewing, (e) theme definition, and (f) theme naming [42]. First, we thoroughly read the documents to ensure familiarization, with independent coding of the initial themes by each author. To ensure accuracy, each author independently returned to the transcripts to confirm that the identified themes were a true reflection of the salient points [38]. Then, the authors compared findings and reviewed the themes, exploring the similarities and differences. Through discussions, the themes were defined and named.

Deductive themes were identified through the literature review and research questions, while the inductive themes emerged from the data analysis (Table 5). We paid particular attention to the processes related to the PADAA project in Wenchuan County and the donor province (Guangdong Province), the new economic situation in Wenchuan, and the link between the two regions.

**Table 5.** Deductive and inductive themes and subthemes.

| Theme | Subtheme |
| --- | --- |
| PADAA and economic structure | Scientific plan<br>Public infrastructure construction<br>Enhanced technology<br>Capital accumulation |
| PADAA and residents' livelihood | High-level infrastructure<br>Employment creation<br>Income structure |
| Economic recovery and "new normal" | Industrial structure transformation<br>Employment and income<br>Tourism and local business renewal<br>Agriculture improvement<br>Investment |
| Knowledge and government self-adjustment | Advanced ideas and idea conversion<br>Professional exchange<br>Inter-governmental cooperation<br>Institutional reform<br>Opening-up |
| The formation of coordinated governance | Long-term cooperation<br>The participation of social organizations<br>Multi-agency partnerships<br>Investment from other areas |

The strong focus was on qualitative, rather than quantitative approaches, because statistical data are generally reported at the end of the year, and most investments complete at that time. Another studied problem (i.e., PADAA effeteness) involves complex causal chains, which cannot be fully explained by pure quantification.

## 3. Results

The analysis of the primary and secondary data outlined in Section 2 suggests that the economic recovery of Wenchuan County can be divided into three stages. The first three years following the earthquake (i.e., 2008–2010) gave priority to the PADAA process and reconstruction (Section 3.1) and the subsequent five years (2011–2015) focused on reform and inter-governmental cooperation (Section 3.2), with a "new normal" economy forming in the past three years (2016–2018) (Section 3.3).

*3.1. The PADAA and Reconstruction Phase (2008–2010)*

3.1.1. Contribution of Guangdong Province during the Recovery Process

As mentioned in Sections 1 and 2.1, the PADAA process is used to assist areas affected by disasters through the mobilization of internal forces of the party and government system. In the case of the Wenchuan earthquake, Guangdong Province acted as the source of funding and technical assistance due to its high economic capacity (Section 2.1).

According to the document of Guangdong Province's government "Guangdong People's All-out Support for Sichuan Earthquake Relief," the province committed to provide for the disaster-affected areas with all the necessary support. Guangdong Province allocated 1% of its GRP every year during the three-year PADAA process, a sum that amounted to a total of approximately 1.2 billion USD. According to the Wenchuan County 2010 annual report, the investment over the three-year PADAA process was 24.4 times higher than the financial expenditure of Wenchuan County in 2007 (b1-gr2010).

During the PADAA period, Wenchuan residents benefited significantly from the transfer of advanced technology and know-how of officials from the donor provinces. Toshihiko Kuroda, the president of the Asian Development Bank, marveled at the speed of Guangdong's assistance when visiting Wenchuan County, suggesting that "by any international standard, the construction here is the fastest!"

In particular, housing for 20 million people was rebuilt in the approximately 130,000 km$^2$ disaster zone. Some sources suggested that due to "the extensive use of modern building materials and techniques in post-disaster reconstruction, great changes have taken place in people's housing structure and decorative facilities" (a2-NDGC005).

Speedy reconstruction and the use of advanced technology were also observed in many other sectors, including health care, education, and industry. Many expert interviews attested to this speed:

*"Without the help of Guangdong, we could not have stood up so fast, Yuexiu Friendship Hospital* [the name commemorates the friendship between Guangdong Province and Yingxiu Town] *was built in just 10 days, and it became the first comprehensive medical institution built on the ruins of the Wenchuan earthquake with the help of Dongguan City."* (A1-YX020)

*"At the conventional construction speed, projects like Wenchuan No. 1 Middle School need more than 500 days. However, 19 buildings have been completely capped in 73 days. They used high technology, and the machines worked almost 24 hours a day. There were 1600 workers in Guangdong, and 179 nights shifts were added in more than 180 days."* (A4-JG001)

3.1.2. Livelihood and Infrastructure Projects

The PADAA for the Wenchuan earthquake adopted a comprehensive approach to the development of a series of projects, most of which were related to livelihoods. In total, 702 projects were undertaken during the three-year PADAA period, all of which were concluded within 730 days (see Table 6). According to the PADAA official report, more than 80% of the investment was used for housing, education, medical treatment, sanitation, urban and rural drinking water, and other livelihood projects [43].

As Table 6 shows, hundreds of different livelihood- and infrastructure-related projects were undertaken through the PADAA process, contributing to the sustainable development of Wenchuan County (b1-YJBG01). The "Ten People's Livelihood Projects" were undoubtedly the most representative, including 450 km of road networks, 10 agricultural and sideline product markets, and clean and safe water for every building, among others. All of these projects were completed in less than 500 days after the Wenchuan earthquake. According to local residents, Guangdong's paired assistance was instrumental for the development of a "general highway connecting all the villages, and furthermore, connecting every household." (A5-KK002).

**Table 6.** Construction projects and allocated funding during the PADAA period.

| Project Type | Quantity | Funding (Billion USD) |
|---|---|---|
| Early relief materials and construction of boarding house | - | 0.362 |
| Housing projects for urban and rural residents | 32 | 0.368 |
| School education facilities | 27 | 0.012 |
| Medical and health facilities | 34 | 0.004 |
| Social welfare facilities | 2 | 0.081 |
| Cultural and sports facilities | 32 | 0.004 |
| Radio and television facilities | 16 | 0.112 |
| Commercial circulation facilities | 15 | 0.088 |
| Other public service facilities in urban and rural areas | 149 | 0.003 |
| Urban and rural road facilities | 203 | 0.219 |
| Urban and rural water conservancy facilities | 118 | 0.012 |
| Industrial recovery projects | 28 | 0.010 |
| Technical assistance and other projects | 46 | 0.267 |
| Total | 702 | 1.542 |

Source: Southcn.com [44].

At the same time the PADAA process upgraded some critical social infrastructure, improving the services provided prior to the earthquake. For example, the Wenchuan People's Hospital became the most advanced modern general hospital in Aba Prefecture and is now considered a three-level B-rate hospital. Wenchuan No.1 Middle School will be incorporated into the regional education center of Wenchuan in the future, which will improve the living situation and quality of life. According to local residents, the infrastructure developed through the PADAA process improved the services offered by critical sectors such as healthcare:

*"Guangzhou's paired assistance has built the material foundation for the hospital and has become a powerful force for the rapid development of the hospital in the past 9 years."* (A5-WZ005)

*"Nowadays, the school, hospital and other infrastructure in Wenchuan may be better than those in many developed regions."* (A5-YX003)

3.1.3. Economic Recovery Projects

As part of the PADAA, Guangdong Province assisted Wenchuan County in developing a good industrial development plan through the construction of industrial parks (Guangdong–Wenchuan Industrial Park) and investment attractions. This process combined Guangdong's market and industrial advantages with the resources and characteristics of the recipient areas. Enterprises in Guangdong Province were encouraged to participate in rehabilitation and reconstruction efforts following market-oriented procedures.

Agricultural recovery and production were carried out by keeping in mind the local characteristics of the agricultural system, adjusting the structure of the agricultural/rural economy, and allocating and expanding the number of leading agricultural enterprises. For example, support was provided to Wenchuan County to improve kiwifruit and honeysuckle production, by adopting advanced planting technologies and promoting large-scale production. Specifically, approximately 0.5 million USD was invested to support the renovation of the Aba Qiangya Industrial Base, which included repairing the tea production and processing facilities so that the only leading agricultural enterprise of Aba Prefecture could resume production. Additionally, an orchid-based flower garden production facility was developed, diversifying the local agricultural economy.

Approximately 10.1 million USD was invested in Xuankou Town, Shuimu Town, and Sanjiang Town in Wenchuan County to build an eco-agricultural circular tourism economic belt to restore and develop local tourism options and to vigorously develop the cultural tourism industry featuring the ancient Qiang ("羌族") culture and the earthquake epicenter at Yingxiu. Some of the main cultural protection and tourism development projects include the Chanshou Street project in Shuimo Town,

the Huangniqiao group project in Buwa Village, Weizhou Town, the memorial site of the original bell tower of Aba Teachers College, the restoration and reconstruction of Luopuzhai Village in Yanmen Town, the Tibetan Village in Shuixiang Town, the ancient city wall in Mianqi Town, Qiangrengu Valley in Longxi Town, and the earthquake memorial system in Yingxiu Town.

Currently, the agricultural (first industry sector) and the service sector (third industry sector) play an important role in the economy of Wenchuan, with household incomes improving dramatically following the reconstruction efforts. While expert interviews and secondary data from the Wenchuan survey team of the National Bureau of Statistics suggest that household income during the PADAA period mainly came from labor related to reconstruction activities, agricultural and tourism income constituted a large proportion of the income after the PADAA process. This indicates the major role that PADAA played in helping the economy both in the short- and long-term. Instrumental to this success was the strategy of Bingli Village, which was eventually adopted by the entire Wenchuan County:

> "*Liwan District paired assistance with Bingli Village . . . After the investigation, Liwan District helped Bingli Village establish an industrial development strategy and develop the first industry and third industry, which is adapting to local conditions . . . "* (A1-WZ001)

### 3.1.4. The Role of the Advanced Working Group Approach

In total, 115 officials from Guangdong were sent to Wenchuan to carry out the projects mentioned in Sections 3.1.1–3.1.3. All of these officials were carefully chosen by the central government. For example, 58 Guangdong officials were selected from more than 5000 people who were dispatched to Wenchuan on August 7, 2008. They were all capable and passionate and mirrored the perspective of the administrative level. Additionally, the leader of the Guangdong PADAA team was from the prefecture government level, which is two levels above that of the leader of Wenchuan County. Therefore, the PADAA team and its leadership had experience and were of a high quality.

During the PADAA process, the Guangdong and Wenchuan teams experienced some of the differences between the work cultures of East and West China, particularly relating to time management and efficiency. For example, in Wenchuan two popular phrases among officials, "马" (which means "immediately") and "恼火得很" (which means "in a hurry" or "unhappy") stopped being used following the arrival of Guangdong officials. Such changes in the working culture improved the performance of Wenchuan officials, improving their efficiency and the respect they commanded from the local communities. These aspects are conveyed by the following interview quotations:

> "马上" *means 'immediately' in Guangdong, but it means a vague period of time in Wenchuan. Lifu Zhen (one official from Guangdong) made a serious and humorous request to officials that changed* "马上" *to a specific time. This case shows the high sense of responsibility and the down-to-earth style of work by Guangdong officials."* (A1-CP007)

> "*Guangdong's support for Wenchuan consisted not only of material and hardware but also of spiritual and ideological measures as well as an advanced market economy management approach and experience in social management."* (A1-WC003)

### 3.2. Inter-Governmental Cooperation and Reform Phase (2011–2015)

#### 3.2.1. Inter-Governmental Cooperation after the PADAA

Following the completion of the PADAA process, Guangdong Province and Wenchuan County agreed to engage in long-term cooperation in the medium term. As part of this commitment, they signed the "Guangdong Sichuan economic and social strategic cooperation agreement" (2010), and "The long-term cooperation framework agreement of Guangdong and Wenchuan" (2010). These agreements focused on technical assistance, management assistance, industrial cooperation, and training of officials, among other areas. They also signed other special agreements relating to cooperation in labor services and tourism. Furthermore, Xinxing County of the Guangdong Province government and the Wenchuan

County government signed a regional long-term cooperation framework agreement in 2013 (a1-ABZ009). These agreements between Guangdong Province and Wenchuan County established a long-term effective cooperation mechanism (a1-CPC017).

From the perspective of Wenchuan County, this "interaction" focuses on reporting and thanking the donor for the performed activities. On the other hand, the Guangdong government is still sending officials, conducting return visits, and providing donations. More importantly, due to this agreement, professional exchanges and training are now frequent. Thirteen Wenchuan officials were sent to Guangdong Province as titular cadres in 2013 (a1-RMRB201301).

It is important to note though that this cooperation goes beyond government agencies. For example, many industrial units in Guangdong maintain regular cooperation with similar industries in Wenchuan. For example, some of Guangdong's enterprises such as Huawei, BYD Co., and Evergrande Group invested in Sichuan (a2-FH20181016). In addition, there is also technical cooperation between education units and social organizations. A classic example is the cooperation between the educational institutions in Wenchuan and Guangdong, such as when 50 teachers were sent for training to Guangdong Province in 2014 (a2-GK201401). Such practices are common in other health-related fields, such as social work, technical support, and guidance for dermatosis prevention (provided by the Dermatology Prevention and Treatment Institute of Guangzhou).

### 3.2.2. Reform of the Wenchuan Government

Wenchuan used to be the most industrialized city in Aba Prefecture and contained many high-polluting manufacturing and extractive industry facilities. However, as most industrial activities were disrupted, industrial output decreased immediately after the earthquake. One of the main outcomes of the PADAA process, and especially of advanced technology transfer, was the development of the Guangdong–Wenchuan Industrial Park. One respondent suggested that the earthquake "was a calamity but also a chance for industrial upgrading", as it helped "eliminate the dirty and high-energy industries." (A1-JX004).

Wenchuan County government further improved its opening up and cooperation after the PADAA process through two main actions: (a) the establishment of the bureau of investment and an exposition, and (b) the participation and representation of Wenchuan County in more expositions. Such examples include the 14th China Western Exposition and the Guangdong–Wenchuan investment explanation for the cooperation of backup and construction in Wenchuan.

During the reconstruction phase the construction of infrastructure was managed through the civil administration department, with public services transferred to other relevant government departments. With the support of the working group of the PADAA, 0.61 million USD was used to establish the Datong Social Worker Station (which is the first social worker organization in Wenchuan) in Wenchuan in December 2009. The contract expired in 2013, and the government of Wenchuan County decided to allow three social work organizations, including Datong, to buy services worth 100,000 USD per year starting in 2014 (A2-DT007).

### *3.3. Formation of the "New Normal" Economy (2017–onwards)*

### 3.3.1. New Economic Structure

As outlined in Section 3.1.3, the economic system of Wenchuan changed rapidly. The earthquake had a strong negative economic effect in its immediate aftermath through the disruption of the local economy, infrastructure, and society in more than ten counties [5]; but the economy rapidly rebounded (Figure 3). Indeed, the GRP recovered to its pre-disaster state within two years, and it grew rapidly until 2014 and has maintained a relatively steady trajectory since then.

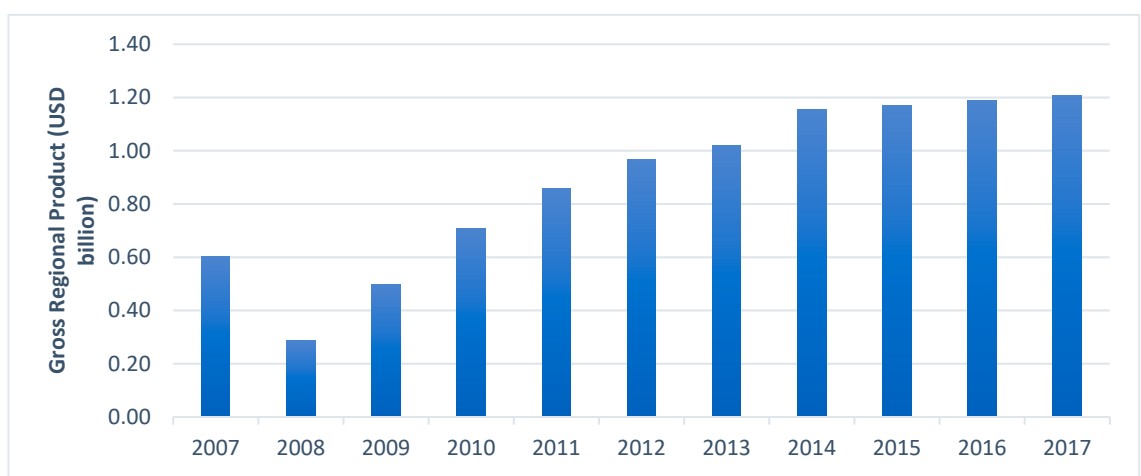

**Figure 3.** Gross regional product of Wenchuan County (2007–2017) (billion USD).

In particular, the industrial sector suffered a devastating blow, with direct economic losses amounting to approximately 10 billion USD. However, since then, the industrial sector has undergone major transformation and development. In 2017, industrial GRP reached 0.8 billion USD, almost four times that in 2008. The service sector also made great progress, and the economic foundation was further consolidated. The fractions of the different sectors in the GRP (i.e., agriculture: industry: services) have become more balanced: from 6.3:77.1:16.6 in 2007, to 6.4:64.3:29.3 in 2017(a1-ZFGZ2007, a1- ZFGZ2017).

The added value of the agricultural sector in 2017 was 3.2 times that in 2008. The data show that the output value of sweet cherries, crisp plums, and fragrant apricots (three of the most famous fruits in Wenchuan) reached approximately 72 million USD in 2017, supporting the income of approximately 70% of the rural residents (a1-ZFBG201711). The vice chairman of the China Post in Wenchuan claimed that Guangdong Province ranked first in parcel deliveries between Wenchuan and all provinces in China except Sichuan: *"Once our special products (including cherries, plums, and apricots) were launched, people often sent them to people who helped them after the earthquake . . . and they often tried to sell their friends special produce from Wenchuan." (A5-WZ012)*

According to the Wenchuan government annual report, the added value of the industrial sector was 0.42 billion USD in 2011 and 0.55 billion USD in 2017. Today, industries in advanced and new technologies, including lithium battery and electric-power industries, play an important role in the industry of Wenchuan (a1-ZFGZ2011, a1- ZFGZ2017).

The service sector plays an increasingly important role, particularly in tourism. In 2017, the county received 6.015 million tourists, 5.75 times greater than in 2007 before the earthquake. This corresponds to an average annual increase of 20%, with the total tourism revenue reaching 0.39 billion USD.

Furthermore, the disposable income of urban residents in Wenchuan County increased substantially and steadily between 2007 and 2017 (Figure 4). However, the disposable income is still lower than that of Sichuan and China, because Wenchuan had high poverty rates until 2018. In fact, the incidence of poverty dropped from 6.8% before the earthquake to currently 0.65%.

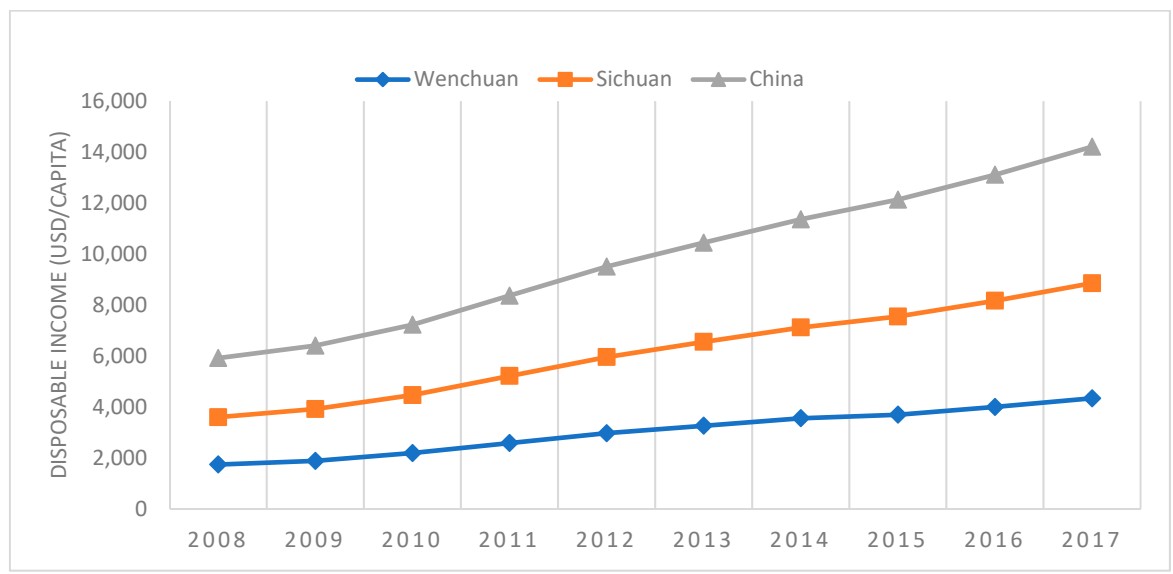

**Figure 4.** Disposable income of urban residents in Wenchuan County, Sichuan, and China (2007–2017) (USD/capita).

### 3.3.2. Investment Attraction and Coordinated Governance

Following the PADAA project and subsequent Wenchuan government reform, increasing investments were attracted and substantial social capital was formed in Wenchuan. Currently, investments from Guangdong to Wenchuan are rather regular reaching 0.48 billion USD in 2018 (a1-WC045). The Guangdong Social Organization Administration played an important role in this activity, with many other organizations building long-term relationships (e.g., the General Charity of Guangdong Province, Sichuan Chambers of Guangdong Province, Guangdong Food (Medicine) Industry Association, Guangdong Food Industry Association, Guangdong Self-Driving Tourism Association, and Guangdong Pension Service Association). According to an official from the Guangdong Social Organization Administration: "We will encourage the association of relevant businessmen of our province and enterprises to visit and invest in Wenchuan." (A1-SHZZ013).

The improved openness further helped promote the increasing formation of ties between social organizations and enterprises towards Wenchuan County. For instance, Wenchuan successfully signed an investment project of 15.94 million USD for "Sichuan Travel of Famous Chinese and Foreign Enterprises" in 2017. According to one respondent: "Due to the improvement of investment environment, more and more people come to Wenchuan, and the housing price in Wenchuan is higher than that in other areas" (C1-XJ201810).

## 4. Discussion

Wenchuan's economic recovery shows strong signs of success following the PADAA process. The rapid recovery and economic transformation towards sustainability is evident in at least five major areas: (a) sustained GRP growth, (b) income growth, (c) employment diversification, (d) diversification and optimization of the industrial sector, and (e) opening up of the economy and ability to attract investments. All of these are strong indications of economic sustainability that were, in one way, or another, fostered by the PADAA process (Section 3). Based on the results mentioned in Section 3, we will (a) discuss some of the key factors underlying the positive economic sustainability outcomes of the PADAA process, and contrast them with recovery experiences from other parts of the world (Sections 4.1–4.4), (b) identify some of the negative outcomes of the PADAA process (Section 4.5), and (c) identify some salient points that local governments can consider during the reconstruction and recovery processes following natural disasters (Section 4.5).

*4.1. Differences with Other Paired Aid Recovery and Reconstruction Processes*

Compared to other paired aid systems, the investment during the Wenchuan PADAA process was much larger and followed a slightly different approach. As discussed below, we identify two important differences with other paired aid systems: (a) the strong top-down effort to mobilize and coordinate support, and (b) the level of donations and the characteristics of the donors.

The paired assistance process during the East Japan earthquake was a "bottom-up" activity, and the government in Japan did not take command [33]. In contrast, the Wenchuan PADAA project was a "top-down" policy in which national agencies were mobilized through "political order". Through this top-down approach the donor provinces were more willing to assist the disaster-affected areas under the political pressure of the central government. The ability to mobilize quickly and effectively allocate resources are crucial for disaster recovery [45], and this was seen as a crucial national mission during the PADAA process. As outlined in Section 3.1.1, the Wenchuan PADAA process attracted international attention due to its rapidity in terms of mobilizing resources and expediting the reconstruction process.

When it comes to the type of donors and the level of donations, the Wenchuan PADAA established greater support from more powerful donors. For example, the support from donors in the L'Aquila area of Southern Italy after the 2009 earthquake was limited and was partly provided by the Province of Trento and the Autonomous Region of Friuli–Venezia Giulia [32]. During the paired assistance for the Japanese Great Hanshin-Awaji earthquake, aid intensity and capacity were also quite limited with regard to dispatching staff and monetary support from paired areas [46]. The donors in Wenchuan PADAA came from a highly-developed region within China, with a much higher administrative level than the disaster-affected areas. This strongly implies that donors in Wenchuan PADAA were much more powerful, richer, and had more advanced ideas than the local governments in the disaster-affected areas. Therefore, these donor provinces were better able to invest at a much larger scale. For example, there are 119 county-level administrative regions in Guangdong (the same as Wenchuan); the GRPs of the 19 provinces that participated in the PADAA were among the highest in China and were individually much higher than those in Wenchuan.

*4.2. Role of Infrastructure Development and Industrial Transformation for Economic Sustainability*

As discussed in Sections 2.1 and 3.1, the earthquake destroyed much of the infrastructure in the disaster-affected areas. Quickly and effectively rebuilding affected infrastructure is essential for enabling successful recovery after major disasters [47]. National government agencies and some types of multi-type aid providers, such as social organizations, social workers, education sector, among others (reconstructing infrastructure is not within the purview of most aid agencies, although they may contribute to it, and international governments and multilateral aid agencies usually do not focus on infrastructure reconstruction in disaster-affected areas [28]), are more likely to respond quickly and effectively to infrastructure needs in disaster-affected areas [48]. During the PADAA process, infrastructure reconstruction was not only carried out quickly (Section 3.1), but also through the cooperation between local governments.

Through this large-scale infrastructure development, the PADAA reshaped the economic structure of the disaster-affected areas, as evidenced by the rapid change of the economic landscape post-PADAA. The comprehensive assistance in multiple sectors and the adjustments and upgrade of the industrial base represented a crucial aspect of the economic rebound and helped facilitate the economic sustainability of Wenchuan.

This "big push" through massive public-led investment [45] was used to overcome the post-disaster financial gaps and enabled the emergence of a new economy to generate self-sustaining growth. Essentially, the Wenchuan economy did not exhibit a downward trend after the end of the three-year assisted reconstruction process, thus avoiding the "forgotten phase" (Section 1). In contrast, the economy has maintained a sustainable growth through infrastructure development and industrial transformation, which is a major strength to attract further investments (Section 3.3.2). For example, the Wenchuan government established a special institute to improve openness and to attract investment in other areas.

However, overly ambitious goals and a lack of effective implementation can lead to incomplete reconstruction, missed development opportunities, and long-term unemployment [49]. A sustainable recovery process "depends on reviving and expanding private economic activity and employment and securing diverse livelihood opportunities for affected populations" [50]. This can be achieved through many initiatives that go beyond the reconstruction of hard infrastructure, such as cash-for-work programs and providing job opportunities [51], skills-training programs [52], owner-building schemes [53,54], and arranging financial help and grants for small businesses and microenterprise schemes [55]. The PADAA processes enabled a series of such measures. In particular, the Wenchuan government turned its attention to agriculture and tourism, which reflected the suggestion of the Guangdong Province. The changes in the agriculture and tourism sector provided increasing numbers of jobs and income to rural residents and citizens.

It is worth mentioning that these structural changes required the Wenchuan government to improve its knowledge base and management capabilities. This was achieved through the close interaction with officials from the Guangdong Province during reconstruction work, the interchange of officials between the two regions, and the training of officials. The exchange, training, and technical guidance of officials in the post-PADAA period are good approaches towards further developing human resources. In this respect, the PADAA donors not only assisted with the actual recovery, but also acted as development teachers.

Overall the restructuring of the industrial base, the formation of capital, the progress of science and technology, the education of bureaucrats, and the improvement of management capabilities all had large positive effects on economic growth and regional development.

*4.3. From Bureaucratic Mobilization to Coordinated Governance Formation*

As outlined above, the state played a much more prominent role during the Wenchuan earthquake reconstruction process. Even though the private sector and Non-Governmental Organizations played a relatively smaller role, their contribution was also important for the success of the reconstruction process. For example, the Wenchuan Datong Social Worker Station provided psychological counseling to residents in disaster-affected areas after the earthquake.

Essentially, this broader participation during the post-disaster reconstruction process gave rise to the local experience of "government-led, market operation and social participation". The meaning of the term "led" is two-fold in this context. On the one hand, it refers to the social forces, that through a cooperation framework, work jointly towards the reconstruction of affected areas under strong motivation of the central government. On the other hand, it refers to a broader societal mobilization as evidenced through the growth of civil society organizations, the strengthening of the public consciousness, and the social responsibility of enterprises.

In this sense, PADAA processes are not simply recovery programs but also productive relationships between two regions, and their respective local governments. In fact, many social organizations and people are mobilized during such recovery processes. The participation of social actors, such as the civil society and enterprises, changed some aspects of local governance that used to involve only government agencies (e.g., through the cooperation of Wenchuan government and the social organizations of Guangdong Province) (Section 3.3.2).

This transformation embodies the logic of coordinated governance that starts from the mobilization of the central government bureaucracy to the eventual mobilization of broader societal actors [56]. Furthermore, through this "state-led corporatism," broader social forces are integrated into the process of reconstruction and recovery, which bridges the possible gap caused by the weakened political pressure [57]. Such broader, effective, and sustained governance actions can play an important role in overcoming the "forgotten phase" after natural disasters [58].

### 4.4. Challenges and Limitations of the PADAA Process

Despite its substantial benefits and success, the PADAA process also faced some challenges and limitations. A better understanding can provide valuable lessons for similar processes in the future. First, to some extent, the PADAA process could be seen as a contract responsibility system for post-disaster reconstruction under the arrangement of the central government. However, due to time constraints, the extent of tasks at hand, and imperfect design during the early stages of the reconstruction, some negative outcomes were visible after PADAA's completion. Such an example was the rebuilding of the new school using technology that exceeds local economic abilities. To ensure the normal operation of the school, additional facilities were needed, which increased maintenance costs later.

Second, even though the economy and the local communities benefitted from massive investment and construction in such a short period, it is also important to effectively maintain infrastructure to sustain economic growth and reduce poverty in the long-term [58]. However, due to the urgent need for infrastructure development, it is not clear whether the rapid and large-scale reconstruction was (a) adequately informed (i.e., some decisions almost needed to be made on the spot), (b) cost-effective, or (c) not environmentally harmful.

Thirdly, while cooperation is the central element of the PADAA system, there were instances of contestation and a lack of cooperation, as organizations sometimes apparently pursued different goals. For example, donors wanted to complete the task as soon as possible, however, the local government had to balance the speed and quality of the recovery process. There were also "culture shocks" between the cooperating officials, considering the different processes followed at the local government level. This led, in some cases, to unpleasant cooperation and conflict between agencies of the Guangzhou and Wenchuan governments (Section 3.1.4).

Fourth, despite the strong cooperation between different actors, there was a relative absence of citizen participation. The Wenchuan recovery was, in a sense, very strong in the way that the government mobilized, but extremely weak in community participation [32]. From a Western perspective, this case is virtually opposite to the philosophy and components of "development recovery," which emphasize the participation of disaster survivors in decision-making to create local "ownership" [32].

### 4.5. Policy Recommendations for Local Governments

The results suggest that the economic recovery of Wenchuan County can be divided into three stages: (a) PADAA process and reconstruction (Section 3.1); (b) reform and inter-governmental cooperation (Section 3.2); and (c) formation of a "new normal" economy (Section 3.3). Even though the outcomes of this research suggest the success of the PADAA process in many aspects (Sections 4.1–4.3), there were also some negative outcomes (Section 4.4). Our study identifies four important aspects that local governments need to consider when engaging in disaster reconstruction through paired assistance programs in China and elsewhere.

First, there is a need to follow a systematic approach towards achieving sustainable economic recovery. Figure 5 contains a conceptual framework that contains the main lessons learned through the Wenchuan PADAA and identifies priority areas for local governments to focus their efforts during disaster reconstruction and recovery. Reconstruction and recovery in Wenchuan required a large number of investments in public infrastructure that would essentially reshape local livelihoods and economic structure in the disaster-affected areas. In this context, knowledge acquisition and openness through institutional reform eased the effort of meeting economic and social development targets. Through the reconstruction and the improvement of the government's economic management ability, a sustainable industrial structure was developed.

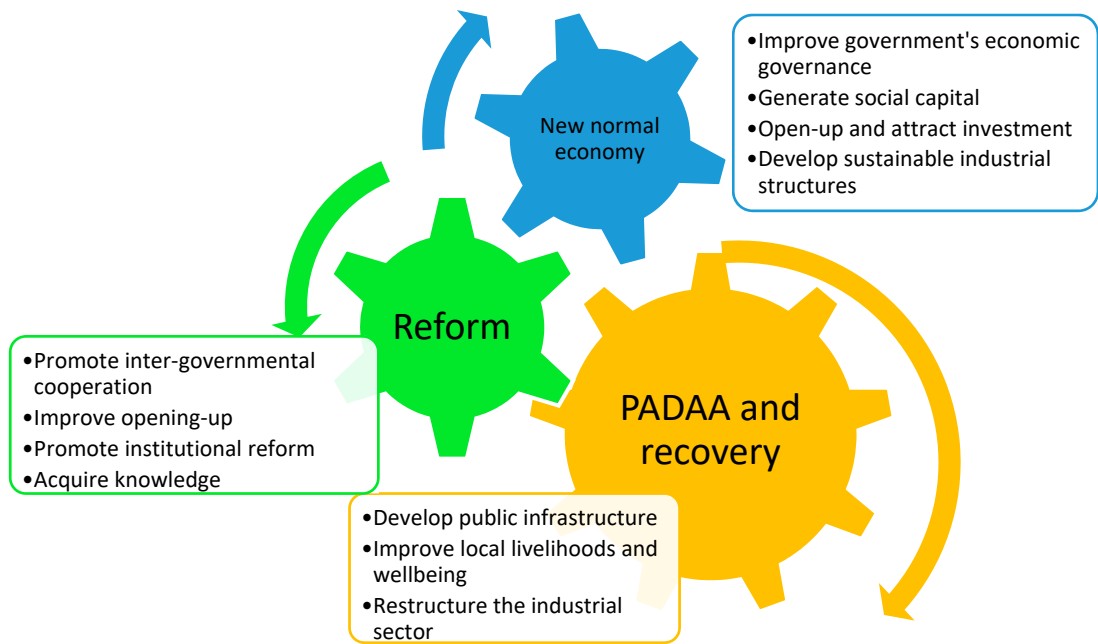

**Figure 5.** Framework of sustainable economic recovery after natural disasters.

Second, the strong role and political order brought by the national government can be a double-edged sword. On the one hand, it helps local governments focus on priority tasks and ignore everything else, as the Chinese central government pushed local governments finish the recovery tasks within three years. On the other hand, under this political order donor governments paid less attention in the needs and requirements of local residents, instead focusing on formally finishing their tasks. In this regard, it is possible that if the central government had more ways to evaluate and supervise PADAA achievements and the overall process, then the outcomes would have been better for the local community.

Third, the disaster recovery and reconstruction processes seemed to have some impact on the local environment. Local governments should pay more attention to preventing and/or reducing the environmental impacts of reconstruction efforts, rather than solely focusing on economic recovery and social development.

Fourth, the PADAA process, in a way, expands the channels for social participation in disaster reconstruction and recovery. As local residents in the disaster-affected areas are the main beneficiaries of reconstruction, it is important to instill the notion that aid workers are helping them rather than working instead of them. Thus, local governments should seek avenues to forge closer participation of local residents in the reconstruction efforts.

## 5. Conclusions

This study used a qualitative research approach to explore the different phases of the Wenchuan PADAA process, and the factors/mechanisms that enabled its ability to catalyze economic sustainability shortly after the Wenchuan earthquake. Overall the Wenchuan PADAA can be characterized as a success story of economic recovery through the successful implementation of a paired assistance program between local governments. Such paired assistance systems might be a solution to disaster recovery and reconstruction, especially in the face of increasing natural disasters in the future. There are at least three important aspects in the Wenchuan PADAA that catalyzed its success: (a) the power of the mobilized national bureaucracy, acting in a "top-down" manner under strong political pressure; (b) the role of donors from developed regions that acted not only as recovery helpers, but also as development teachers; and (c) the wide scope of the process that contained a series of projects (mostly related to livelihoods), that achieved large capital infusion and rapid infrastructure construction.

The PADAA process improved economic sustainability following a systematic approach. First, the PADAA facilitated a large number of investments in public infrastructure, leading to the restructuring of economic activity and livelihoods in the disaster-affected areas. Secondly, it catalyzed knowledge acquisition and openness through institutional reform, thus easing the efforts of meeting economic and social development targets. Through the reconstruction and the improvement of the Wenchuan government's economic management ability, a sustainable industrial structure was developed.

**Author Contributions:** Conceptualization, Z.W.; methodology, Z.W. and X.Z.; software, X.Z.; validation, Z.W. and X.Z.; formal analysis, Z.W. and X.Z.; investigation, Z.W. and X.Z.; resources, Z.W. and X.Z.; data curation, Z.W. and X.Z.; writing—original draft preparation, X.Z.; writing—review and editing, Z.W. and X.Z.; visualization, X.Z.; supervision, Z.W.; project administration, Z.W.; funding acquisition, Z.W.

**Funding:** This work was supported by the Key Projects of Philosophy and Social Sciences Research of the Ministry of Education (NO. 16JZD026) and the Key Projects of the National Social Science Fund (NO. 15AZZ002).

**Acknowledgments:** This work was made possible by contributions from residents in Wenchuan earthquake disaster-affected areas. The residents and officials provided significant help in the questionnaire survey. The authors express their sincere appreciation for their generous cooperation and valuable input. The authors would like to thank David Alexander, Haibo Zhang, and the anonymous reviewers for their constructive comments regarding this article.

**Conflicts of Interest:** The authors declare no conflicts of interest.

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
