# Peer review of "How Does Paired Assistance to Disaster-Affected Areas (PADAA) Contribute to Economic Sustainability? A Qualitative Analysis of Wenchuan County"

_sustainability, doi:10.3390/su11143915_

Round 1
Reviewer 1 Report
Overall comments:
Title: How Does Economic Recovery Turn to Sustainable After the Earthquake? – A Qualitative Case Study
The paper is reasonably well written and structured - although the final version of the text does need to be edited by a native English speaker to take out spelling errors, grammar and syntax.
The manuscript needs language consistency throughout the paper.
Abstract
Good Abstract.
Introduction
The introduction is good and, however, it can be expandable with more recent global examples.
Table one is not in good format. Please revise.
Fig. 1 caption should not on the top the figure.
Should be the bellow of figure. The numbers in this figure are not clearly visible.
2.PADAA and Wenchuan’s Economic Recovery
There is no need to give the currency in Chinese terms. Already authors gave the information in $.
That is enough to the international readers.
Methodology
Straightforward methodology.
Results
I was confused with the results section. Please revise.
Discussion
Good
Conclusion
The conclusion needs revisions. It should be in one paragraph.
References
Some references are very old Please update the references with the latest references.
Author Response
Response to Reviewer 1 Comments
Thanks for your comments. We appreciate all your suggestions, they helped a lot. We made a great revision, and welcome new comments.
Point 1: The introduction is good and, however, it can be expandable with more recent global examples. Table one is not in good format. Please revise. Fig. 1 caption should not on the top the figure. Should be the bellow of figure. The numbers in this figure are not clearly visible.
There is no need to give the currency in Chinese terms. Already authors gave the information in $. That is enough to the international readers.
Response 1: We appreciate the comments, and made revision according to the suggestions.
Point 2: Straightforward methodology.
Response 2: We made revision, please see the new version.
Point 2: I was confused with the results section. Please revise.
Response 2: Acoording to the “Research Approach” in the revised version, we tried to reconstruct the event. The economic recovery in Wenchuan can be divided into three stages. The first three years (2008-2010) gave priority to PADAA and reconstruction; the second five years (2011-2015) gave priority to self-promotion and inter-governmental cooperation; the “new normal” economy was formed in the last three years (2016-2018).
Point 3: The conclusion needs revisions. It should be in one paragraph.
Response 3: We made revision, however, we prefer to using three paragraphs to express our conclusion.
Point 4: Some references are very old Please update the references with the latest references.
Response 4: We made revision, please see the new version.
Reviewer 2 Report
1. The citation of the authors should be according the requirements.
2. In line 40: “These numbers reflect the long and arduous road to economic recovery”. It need more justification of this sentence from the economic point of view. The decrease of GDP does not reflect the main problems of economic situation.
3. More scientific discussion should be about the topic: economic recovery after the earthquake; (line 4); Impact of it to economy; the recovery cases from other regions;
4. The design of the article should be improved. The goal of the article, the actuality, the novelty and level of scientific research of the analysed problem would strengthen the introduction part. It’s not clear, what is scientific problem of the article. The methodology part need the scientific discussion about these concepts: recovery and economic recovery, sustainable recovery; what are they and what are differences of them; What is resilience of region?
5. As the authors analyze economic recovery, the modeling of economic recovery based on the analysed case would strengthen the article.
Author Response
Response to Reviewer 2 Comments
Thanks for your comments. We appreciate all your suggestions, they helped a lot. We made a great revision, and welcome new comments.
Point 1: The citation of the authors should be according the requirements.
Response 1: We made revision.
Point 2: In line 40: “These numbers reflect the long and arduous road to economic recovery”. It need more justification of this sentence from the economic point of view. The decrease of GDP does not reflect the main problems of economic situation.
Response 2: We deleted this sentence, however, we provided more details economic recovery.
Point 3: More scientific discussion should be about the topic: economic recovery after the earthquake; (line 4); Impact of it to economy; the recovery cases from other regions;
Response 3: We provided more details about economic recovery after the earthquake and impact of it to economy.We cited more to compare with the approaches similar to PADAA have been used in Japanese Great Hanshin-Awaji Earthquake, Tohoku earthquake recovery, Italian earthquakes, and more cases.
Point 4: The design of the article should be improved. The goal of the article, the actuality, the novelty and level of scientific research of the analysed problem would strengthen the introduction part. It’s not clear, what is scientific problem of the article. The methodology part need the scientific discussion about these concepts: recovery and economic recovery, sustainable recovery; what are they and what are differences of them; What is resilience of region?
Response 4: We made the scientific problem clearer, we focused on the topic “How Does Economic Recovery Turn to Sustainable After the Earthquake?” To fill this knowledge gap, this article uses a qualitative research method to explore the factors and mechanisms that enable the PADAA to help disaster-impacted areas to overcome the “forgotten phase” and display characteristics of sustainability. We improved the design of the article. First, we tried to reconstruct the event base on the data that we collected from three phases. Second, we used thematic analysis to explore the factors enable to help Wenchuan’s economy turn to sustainable. Third, we combined the two parts to discuss the mechanism of economic sustainability after the Wenchuan earthquake. Fourth, we tried to propose a theoretical framework for external assistance and economic recovery.
And we also discuss recovery and economic recovery, sustainable recovery in the introduction part.
Point 5: As the authors analyze economic recovery, the modeling of economic recovery based on the analysed case would strengthen the article.
Response 5: We tried to propose a theoretical framework for external assistance and economic sustainability after a disaster.

Round 2
Reviewer 1 Report
The authors have addressed my concerns.
Author Response
Thank you very much for your comments.
It helps a lot.
I will keep on revising my manuscript.
Wish you have a good day.
Reviewer 2 Report
What is the purpose of the article. It’s not clear, what the authors want to analyse – how to turn the Economic Recovery to Sustainable recovery or the factors and mechanisms that enabled PADAA to act. There’s the mismatch between the name of the article and the content.
Also there is the need of clarity in the concepts used in the article: what are the differences among sustainability, economic sustainability (line 16), sustainable economic development (line 46). What are the interaction between these concepts and with the concepts of sustainable recovery.
The authors stated only good aspects or tendencies of PADAA to economic sustainability. But the economic justification of the of PADAA impact to sustainability or economic sustainability is still lack (more justification would provide the reliability; the statements are not enough for scientific article) (based on the object of the analysis and definitions of the concepts).
It’s still not clear, what is the novelty of the article. What is the implication of the article to science.
Author Response
Response to Reviewer 2 Comments(round 2)
Dear Reviewer,
On behalf of my co-authors, we thank you very much for giving us an opportunity to revise our manuscript, we appreciate editor and reviewers very much for their positive and constructive comments and suggestions on our manuscript entitled “How Does Economic Recovery Turn to Sustainable After the Earthquake? – A Qualitative Case Study”. (ID: sustainability-454839).
We have studied the reviewer’s comments carefully and have made revision which marked in red in the paper. We have tried our best to revise our manuscript according to the comments. Attached please find the revised version, which we would like to submit for your kind consideration.
We would like to express our great appreciation to you and reviewers for comments on our paper. Looking forward to hearing from you.
Yours sincerely,
Xiaojun
Here are the response to the comments below.
Point 1: What is the purpose of the article. It’s not clear, what the authors want to analyse – how to turn the Economic Recovery to Sustainable recovery or the factors and mechanisms that enabled PADAA to act. There’s the mismatch between the name of the article and the content.
Response 1: It’s a great comment, it helps me to focus more. I also confused before, however, I know I should give up the name which seems more interesting now. From my article, I tried to explore the factors and mechanisms that enable the PADAA to help disaster-impacted areas to overcome the “forgotten phase” and display characteristics of sustainability. So, maybe the name of the article now “How Does Paired Assistance to Disaster-Affected Areas Project Help Wenchuan Economy Turn to Sustainable? – A Qualitative Case Study” will be more suitable. What do you think? Looking forward to your further comments.
Point 2: Also there is the need of clarity in the concepts used in the article: what are the differences among sustainability, economic sustainability (line 16), sustainable economic development (line 46). What are the interaction between these concepts and with the concepts of sustainable recovery.
Response 2: Sorry, the problem was that I made it too complicated.
I give up “sustainable economic development” and try to explain “sustainability” and “economic sustainability” bellow (please see the second paragraph):
Sustainability implies the capacity to continue an activity or process indefinitely [5], therefore, the concept of sustainability has been adopted by hazards researchers and applied to recovery[6][7][8]. Sustainability is confronted with deeply rooted social, economic and environmental [9][10]. It is essential, not only to cope with the impacts but also to help ensure that the region sustains its economic growth. Mileti (1999) have defined sustainability in the context of hazards and disasters, "sustainability means that a locality can tolerate—and overcome—damage, diminished productivity, and reduced quality of life from an extreme event without significant outside assistance" [11]. Based on this definition, economic sustainability is the ability of an economy to support a defined level of economic production indefinitely [12]. Sustainable economic development should be characterized by a reasonable rate of growth of the economy, with natural concern for and preservation of the environment, and the good prospects for future growth and development [13].
Point 3: The authors stated only good aspects or tendencies of PADAA to economic sustainability. But the economic justification of the of PADAA impact to sustainability or economic sustainability is still lack (more justification would provide the reliability; the statements are not enough for scientific article) (based on the object of the analysis and definitions of the concepts).
Response 3:
Thanks for your comments, it helps a lot.
We finally focus on the topic “How Does Paired Assistance to Disaster-Affected Areas Project Help Wenchuan Economy Turn to Sustainable?” So, we stated good aspects or tendencies of PADAA to economic sustainability. We also found some bad aspects, just like we wrote in the article, “We cannot sure that quick, big scale reconstruction during the PADAA process is scientific, however, it paid locals to become involved in reconstruction work.”
As for “more justification would provide the reliability; the statements are not enough for scientific article”, it was very difficult to find statistic data to answer the question. Actually, we have tried that, for example, panel data analysis on 39 disaster-impacted areas, however, we failed. The most important reason is that statistical data are generally based on the end of the year, and most investments were complete at that time. To answer this important question (which we think) we decided to use a qualitative case study. In this qualitative research, the single case study method can be applied to explore the problems listed above, and the method has obvious advantages in terms of investigating new phenomena and dynamic processes. Deductive themes can be generated from the research aims and existing literature, while inductive themes can be generated from the data and the grinding process. Economic recovery after the earthquake is a relative concept and is highly context based, and case and deductive themes analysis is particularly suitable for studying such events.
Of course, we agree that “the statements are not enough for scientific article”. That was why use NVivo 12 software to analyze the interviews and secondary data. We used deductive themes analysis to rebuild the process of Wenchuan economic recovery, which comprises a process of coding in six phases to create established, meaningful patterns. These phases include becoming familiarized with data, generating initial codes, searching for themes amongst codes, reviewing themes, defining themes and naming themes. The findings were confirmed by investigator triangulation and reflexivity.
We totally agree with your comment that analysis should “based on the object of the analysis and definitions of the concepts” and “based on the object of the analysis and definitions of the concepts”. So, we revised the whole discussion section. Based on the results mentioned above, we tried to discuss the economic justification of PADAA impact on sustainability, with an emphasis on economic structure, residents’ livelihood, knowledge acquisition, and capital substitution.
Point 4: It’s still not clear, what is the novelty of the article. What is the implication of the article to science.
Response 4:
Basically, there are two aspects of novelty, one is the comparison to the similar approaches, the other one is the attempts to the theoretical framework for PADAA and economic sustainability.
1) The approaches similar to PADAA have been used in the recovery process of the Japanese Great Hanshin-Awaji Earthquake, Tohoku earthquake, and Italian earthquakes, and so on, however, the plans didn’t work well. Conventional research on the PADAA and disaster recovery has mainly focused on the operation mechanisms before the associated programme has finished; thus, little is understood about the economic consequences of the PADAA, especially in terms of what happens after the PADAA. Few scholars have paid attention to the effect of the PADAA on long-term recovery following the PADAA. However, the sustainability of the program is questionable, especially because the large-scale investments required in the assisted areas may not be sustainable.
2)This study provides some important insights into how factors that were outlined in paired assistant and sustainable economic recovery. Based on a thorough review of the literature and expert judgments, using a qualitative research method and NVivo 12 software to explore the factors and mechanisms that enabled PADAA to act on economic sustainability after the Wenchuan earthquake. We tried to propose a theoretical framework for external assistance and economic sustainability after a disaster. Facing disaster, a sustainable economic recovery process should be step-by-step: (1) with PADAA and a large number of investments on public infrastructure, the livelihood and economic structure were reshaped in disaster-impacted areas; (2) knowledge acquisition, self-adjustment, and meet the needs of new economic and social development through institutional reform, and enlarge open-up level; and (3) base on the reconstruction and the improvement of the government's economic governance ability and concept, more and more social capitals were attracted, sustainable industrial structure was built. Through paired assistance system, the government of disaster-affected areas achieved the conversion of ideas, the progress of science, the popularization of education, and improvement of the management level.
Round 3
Reviewer 2 Report
The justification, why the qualitative research was used for the problem analysis is needed. Why other methods were not used.
Usually in the conclusions, the scientists do not present the figures. I recommend to put the picture to other part of the article.
Author Response
Response to Reviewer 2 Comments (round 3)
Dear reviewer,
On behalf of my co-authors, we thank you very much for giving us an opportunity to revise our manuscript, we very much for your positive and constructive comments and suggestions on our manuscript entitled “How Does Paired Assistance to Disaster-Affected Areas (PADAA) Contributed to the Economic Sustainability of Wenchuan Country? – A Qualitative Analysis”. (ID: sustainability-454839).
We have studied your comments carefully and have made revision which marked using the "Track Changes" function in the paper. We have tried our best to revise our manuscript according to the comments, and welcome new comments.
We would like to express our great appreciation to you and reviewers for comments on our paper. Looking forward to hearing from you.
Yours sincerely,
Xiaojun
Responses to the reviewers’ comments
Point 1: The justification, why the qualitative research was used for the problem analysis is needed. Why other methods were not used.
Response 1: We made a revision. Please see the revised revision. “It’s very common to answer an economic question with empirical research and statistical data, however, it’s very difficult to analyze the statistical data in my case, because statistical data are generally based on the end of the year, and most investments were complete at that time. Furthermore, the problem involves more complex causal chains, which cannot be fully explained by pure quantification. So, qualitative research methods were used to investigate the factors and mechanisms that enabled the success of PADAA.”
Point 2: Usually in the conclusions, the scientists do not present the figures. I recommend to put the picture to other part of the article.
Response 2: Good suggestion. We put the picture to “4.6. Policy Recommendations for Local Governments”.